# Fungi and Algae as Sources of Medicinal and Other Biologically Active Compounds: A Review

**DOI:** 10.3390/nu13093178

**Published:** 2021-09-12

**Authors:** Joanna Ślusarczyk, Edyta Adamska, Joanna Czerwik-Marcinkowska

**Affiliations:** 1Institute of Biology, Jan Kochanowski University, 25-420 Kielce, Poland; joanna.slusarczyk@ujk.edu.pl; 2Department of Geobotany and Landscape Planning, Faculty of Biology and Veterinary Sciences, Nicolaus Copernicus University, 87-100 Toruń, Poland; adamska@umk.pl

**Keywords:** algae, fungi, lichens, mushrooms, biologically active compounds, functional food, pharmaceuticals sources

## Abstract

Many species of fungi including lichenized fungi (lichens) and algae have the ability to biosynthesize biologically active compounds. They produce, among others, polysaccharides with anticancer and immunostimulatory properties: (1) Background: This paper presents the characteristics of the most important bioactive compounds produced by fungi and algae; (2) Methods: Based on the example of the selected species of mushrooms, lichens and algae, the therapeutic properties of the secondary metabolites that they produce and the possibilities of their use are presented; (3) Results: The importance of fungi, especially large-fruited mushrooms, lichens and algae, in nature and human life is discussed, in particular, with regard to their use in the pharmaceutical industry and their nutritional value; (4) Conclusions: The natural organisms, such as fungi, lichenized fungi and algae, could be used as supplementary medicine, in the form of pharmaceutical preparations and food sources. Further advanced studies are required on the pharmacological properties and bioactive compounds of these organisms.

## 1. Introduction

Natural compounds are all organic or inorganic chemical compounds that have a relationship with living organisms, i.e., which are produced by or extracted from the environment and processed by such organisms. These include all primary metabolites, secondary metabolites and biomacromolecules occurring in living organisms. Each compound in a living organism has its own rank, biogenesis pathway, performs specific functions in the organism, and has a potential application. Mushrooms and algae play a significant role both in nature and in the human economy. About 700 species of fungi have been found to have therapeutic properties [1], so they may be a good source of biologically active compounds for use in the pharmaceutical industry. Many species of edible mushrooms, lichens and algae are a source of compounds with dietary and medicinal properties, exhibiting anticancer, immunostimulatory (glucans, glycoproteins, sesquiterpenoids, triterpenoids), anti-atherosclerosis (chitin, chitosans, statins), antibacterial and antifungal (antibiotic) as well as antioxidant (sterols, tocopherols, flavonoids, carotenoids, indole compounds and phenolic compounds) properties [2,3,4,5,6]. The healing and antioxidant effect of edible mushrooms, lichens and algae complements their nutritional value. Free radicals are important factors causing many pathological processes in the human body and favor the development of what are commonly known as the diseases of civilization. For this reason, fungi have also been used as functional food, i.e., they are products that have proven beneficial effects on health over and above that resulting from the presence of nutrients traditionally considered essential in them [7,8,9]. Research on the health-promoting properties of fungi and algae is being carried out intensively in many research centres around the world. This interest is in line with the current trend of searching for natural substances with a beneficial effect on the human body. Fungi and algae are a potential source of natural antibiotics and antioxidants that would be safe to use and have no side effects.

This article reviews the latest scientific reports on bioactive selected substances contained in fungi and naturally occurring algae mainly in ecosystems in Europe. Many species of large-fruited mushrooms have long been valued for their taste and aroma. Currently, more and more attention is being paid to their therapeutic properties and the presence of valuable secondary metabolites in them. The therapeutic importance of the *Basidiomycota* groups appreciated in the countries of the Far East, where the tradition of using preparations of mushroom origin is several thousand years old. These mushrooms biosynthesize metabolites with immunomodulating, anticancer, antiviral, antibacterial, antifungal, anti-inflammatory, antioxidant, antidiabetic, neurotonic, hepatoprotective, and blood triglyceride and cholesterol levels and blood pressure lowering properties. Mushrooms such as: *Ganoderma lucidum*, *Hericium erinaceus*, *Lentinus edodes*, *Cordyceps sinensis*, *Sclerotinia sclerotiorum*, *Trametes versicolor*, are consumed as food for human health benefits as functional foods and nutraceuticals [10,11,12,13,14].

An important group of fungi that produce bioactive substances with great medical potential are lichens. Lichens are organisms composed of fungal and photosynthetic partners (green algae or cyanobacteria, or both at the same time). They also consist of bacteria and viruses [15,16]. Lichen species comprise more than 20% of the global fungal biodiversity [17]. Lichens are sources of bioactive substances used as medicine and food. Approximately, 1.050 secondary compounds have been identified. Most of them are characteristic only to lichen-forming fungi [18]. Lichen secondary metabolites, including amino acid derivatives, sugar alcohols, aliphatic acids, macrolytic lactones, monocyclic aromatic compounds, quinines, chromones, xanthones, dibenzofurans, depsides, depsidones, depsones, terpenoids, steroids, carotenoids, and diphenyl ethers indicate the great potential of lichens for pharmaceutical purposes [4,15,17,18]. Unlike medical mushrooms, their slow growth is a problem of the production of these substances in cultures [19].

Algae are also a natural source of bioactive compounds [20,21,22]. Marine species of algae are a well-known and valued raw material containing many bioactive compounds such as carotenoids, fatty acids, polysaccharides, amino acids, tocopherols, flavonoids and others, therefore they are used as constituents in medicines, cosmetics, and dietary and food supplements. The chemical composition of freshwater algae is very poorly understood so far. This study focuses on the health-promoting effect of algae on the human body. The potential use of algae and algal extracts in medicine and the cosmetics industry is discussed. Biologically active substances obtained from marine macroalgae are a valuable raw material for the pharmaceutical and cosmetic industries, so far only used to a small extent [23]. Algae are particularly attractive as natural sources of bioactive molecules, since these algae have the potential to produce these compounds in culture that enables the production of structurally complex molecules, which are difficult or impossible to produce by chemical synthesis.

Worldwide, more than 200 species of marine algae have commercial value and are used commercially. These include, among others, various types of brown giant seaweeds from underwater forests, red algae, often used in nori to wrap sushi, and microscopic blue-green algae, which are a popular ingredient in cosmetic creams. This growing and increasingly diverse need for algae may lead to their overexploitation, potentially resulting in a host of problems. Algae, so-called first-tier producers, play a key role in aquatic ecosystems. A shortage of algae could lead to coastal erosion, loss of biodiversity, and a reduction in water quality. It would also have negative effects on the food chain and marine habitats. Additionally, irresponsible aquaculture of algae may lead to pollution of water and local ecosystems and a decline in wild populations. Independent certification may occur helpful in the building of the fast-growing marine algae industry by promoting sustainable and socially responsible production. The purpose of certification would be to protect marine ecosystems and provide a livelihood for those who depend on the acquisition of algae resources [20,24,25,26,27]. The literature review on the study of natural sources of medicinal and biologically active compounds–nutraceuticals is very extensive. Numerous studies concern not only plants (herbal medicine, use as food) but also fungi, including lichens and algae, bacteria and cyanobacterial [28]. Of particular interest are joint studies on several groups of organisms. In the available databases, we have not been able to find a combined description of fungi (mushrooms), lichens, and algae in one paper, therefore we propose such a review. For example, in Poland, paradoxically, despite the tradition of collecting fungi for food (mycophilia), there is a lack of tradition and awareness of their use for medicinal purposes. Similarly, this is apparent with regard to algae and lichens. Although it is commonly known that algae and lichen come with a great potential for the protection of human health and require further intensive research, the scientists are still lacking data and mycological material on the subject. It is extremely important to raise awareness about the value of these resources. It is very interesting that these organisms used in ethnic medicine for thousands of years are valuable sources of biologically active substances, which is confirmed by contemporary research.

The aim of this study was to characterize the bioactive substances and nutritional value of edible mushrooms, lichenized fungi (lichens) and algae. Particular attention was paid to bioactive substances that determine the human health-promoting properties of individual species, based on the results of the latest scientific research.

## 2. Methodology

An extensive literature search was performed to identify the major bioactive substances and nutritional value of edible mushrooms, lichenized fungi, and algae. Furthermore, this review attempted to summarize the active constituents of fungi/lichens/algae and the compounds occurring in them. Various electronic databases such as Science Direct, PubMed, Taylor & Francis, Wiley, along with Google Scholar search engine were used for the literature survey. Papers with information on biologically active compounds in fungi, lichens and algae, and medicinal properties, have been included. Keywords such as pharmaceutical sources, biologically active compounds, medicinal mushrooms, edible mushrooms, functional foods, algae, fungi, lichens, nutraceuticals, ethnomycology, ethnolichenology, ethnomedicine were used to search the literature resources. Additionally, information on particular species of fungi, lichens and algae was searched for health-promoting properties by entering their Latin names, also in combination with use and importance to people. The articles were also filtered for the presence of biologically active compounds such as phenolic compounds, flavonoid, carotenoid, polysaccharides, indole, sterols and vitamin in fungi, lichens and algae, as well as properties of these compounds such as anticancer, antioxidant, neuroprotective etc., and their nutritional characteristics.

The literature cited in this review paper consists of 189 references, which includes research and review papers. Studies or trials that did not include basic details were excluded. The results were compared, correlated and discussed to help researchers to further explore the possibilities of using the biologically active compounds in other research areas, as well as human health-promoting properties of individual species of mushrooms, lichens and algae.

## 3. Nutritional Value of Edible Mushrooms, Lichens and Algae

The nutritional value of individual mushroom species is determined by their chemical composition. Mushrooms are characterized by a high-water content, which means that their energy value is low and amounts to approx. 50–70 kcal/100 g. The main nutrients found in mushroom-fruiting bodies are proteins, carbohydrates, fats, including EFAs (Essential Fatty Acids), fibre, and vitamins and minerals [8]. Proteins constitute 1.5–3.6% of the fresh weight of fruiting bodies [29] and are characterized by a rich set of amino acids, including the presence of all exogenous amino acids, i.e., lysine, methionine, tryptophan, threonine, valine, leucine, isoleucine, histidine and phenylalanine [30]. The presence of eight essential amino acids has been demonstrated in the fruiting bodies of the cep mushroom (*Boletus edulis*) [31]. The level of amino acids is an indicator of the nutritional value and flavor of mushrooms because amino acids give them taste [32]. Mushroom protein is also highly bio-available. Mushrooms contain 4.7 to 6.9% of carbohydrates, including 2.7 to 3.9% of dietary fibre [7,33]. Dietary fibre is divided into water-insoluble fibre, which is mainly chitin, and soluble fibre, which is dominated by beta-glucans and chitosans [34]. Among carbohydrates there are simple sugars, disaccharides, polysaccharides and sugar-protein complexes. Polysaccharides, which show a strong biological activity, play a special role. Mushroom-fruiting bodies also contain 0.4 to 0.9% fats. These include fatty acids, mono-, di- and triglycerides, as well as sterols and phospholipids. Mushrooms also contain unsaturated fatty acids, including linolenic acid. The content of fatty acids in mushroom-fruiting bodies is low and amounts to 2–8% dry weight. On the other hand, the ratio of polyunsaturated fatty acids to saturated fatty acids is favorable. About 75% of the total fatty acid content is made up of EFAs, of which oleic and linolenic acids are the most important. The main saturated fatty acid is palmitic acid [35]. For example, in the fruiting bodies of the cep mushroom, 30% of the total fat content is made up of polyunsaturated fatty acids [36]. The optimal proportion of unsaturated to saturated fatty acids and the presence of linolenic acid make mushrooms a functional food. The percentage of fatty acids (in 100 g of total fatty acids) in mushrooms varies greatly: linoleic acid ranges from 0.0–81.1%, oleic acid between −1.0 and 60.3%, and linolenic acid from 0.0–28.8%. The consumption of essential fatty acids in balanced proportions (1:1 or 2:1 *n*-6/*n-*3) prevent obesity as an unbalanced *n*-6/*n*-3 ratio has been associated with adipogenesis. Also, essential fatty acids participate in high density lipoprotein formation, which carries fat from the blood to the liver and reduces the risk of cardiovascular disorders [35]. The *n*-3 to *n*-6 ratios are (28.45 ± 7.38% in cultivated edible compared to 16.19 ± 7.38 to 55.42 ± 9.03% in the wild medicinal mushrooms) higher than the recommended value of 2:1 or 3:1 in human diets.

A summary of the content of the most important nutrients in several species of wild mushrooms is given in Table 1.

Mushrooms also contain many minerals [39]. They contain significant amounts of K and P and lower amounts of Ca, Mg and Na. These minerals constitute 56 to 70% of ash, including K approx. 45%. There are also trace elements in fruiting bodies such as Cu, Zn, Fe, Mg, Mo, F, selenium Se, Co and Ti. Differences in the content of minerals occur between species and also between the individual parts of the fruiting body. A higher content of the elements Ag, Al, Cd, Cr, Cs, Cu, Fe, Hg, K, Mg, Mo, Pb, Rb, Se, V, and Zn has been shown in the cap than in the stipe of *Leccinum scabrum* (*Lentinus edodes*) [40]. The shiitake mushroom is a species that is particularly rich in Ca. The presence of trace elements such as Cu, Zn, Fe, Mg, Mo and Se has also been noted in the fruiting bodies of various mushroom species. Mushrooms of the genus *Pleurotus* are superior to other cultivated species in terms of their Cu content. Among wild species, the highest K content is found in the cep mushroom (*Boletus edulis*), brown birch bolete (*Leccinum scabrum*), and chanterelle (*Cantharellus cibarius*) [29]. In fruiting bodies of wild mushrooms, a significant content of B vitamins, especially B6, PP, B2, B1 and H, has been found. The content of vitamin B2 in mushrooms is higher than in vegetables. In addition, mushrooms contain small amounts of vitamin C, B1, as well as trace amounts of vitamin B12, D2 and E [41,42]. Some orange-colored species, such as chanterelles, are a source of provitamin A [43]. Among the cultivated species, the common mushroom, oyster mushroom (*Pleurotus ostreatus*), enoki (*Flammulina velutipes*), and shiitake (*Lentinus edodes*) are particularly rich in B vitamins [44]. Among the wild-growing species with a high content of vitamins, the slippery jack (*Suillus luteus*) stands out [38]. The nutritional value of mushrooms is highest immediately after harvesting, so they should be consumed as soon as possible. During storage, the nutritional value is reduced and there are unfavorable organoleptic changes. Certain nutrients are also reduced during heat treatment. During boiling, some of them pass into the water or are destroyed due to high temperature [45]. The least loss of nutritional value occurs during freezing. Due to the content of valuable components, it is worth including mushrooms in the diet not only in the autumn season. Mushroom preserves, such as dried mushrooms and pickles, can be eaten all year round. However, due to the fact that mushrooms are hard to digest, they should not be given to children, the elderly or those with gastrointestinal diseases. Edible mushrooms are an important raw material in the food industry. According to GEMS/Food data, a statistical EU citizen eats approximately 1.5 kg of mushrooms per year, mainly from cultivated varieties [46,47,48]. Their fruiting bodies are used, among others uses, for the production of delicatessen products, as well as for a garnish for cheese and cold meats. The conditions for placing mushrooms and mushroom products on the market in Poland are regulated by the Act on Food and Nutrition Safety [49].

There has been a tradition of mushroom picking in Poland for centuries. Mushroom collecting is still an important element of the culture [50]. Therefore, Polish society belongs to the so-called mycophiles [51]. Currently, in Poland, mushrooms are used mainly for taste, and very rarely for medicinal purposes. In comparison, the inhabitants of the British Isles are usually presented as an example of mycophobia; however, collection of edible wild mushrooms was practiced locally. In traditional cultures, mushrooms are picked and eaten in European countries [52] and East Asia [53].

Lichenized fungi (lichens) are recommended as a preferred source of medicines or functional foods. Lichens have been used as a food especially in China and Japan, also in India, Nepal, Africa and European countries [54,55,56]. Edible lichens such as *Cladonia gracilis*, *C. stellaris*, *Dermatocarpon miniatum*, *Lobaria pulmonaria*, *Ramalina fastigiata*, *R. sinensis*, *Sulcaria sulcata* and others in the Himalayas and southwestern China are consumed as a vegetable, steamed, fried and made into soup, while *Thamnolia subuliformis*, *T. vermicularis*, among others, are prepared as tea [55,57].

The nutritional value of lichens is mainly reflected in the high content of carbohydrate (53.2–79.08%) and fiber (5.386–16.36%) and low-fat content (1.3–6.5%). Lichens are rich in mineral elements, and good protein sources (5.95–16.2%). The literature data showed the presence of isoleucine, leucine, methionine, phenylalanine, threonine, tryptophan and valine as a result of the amino acid content in lichen thalli *Rimelia reticulate* from Africa [4]. For example, during the war in the territory of Bośnia and Hercegovina, fungi and also lichens were used by humans for food due to poor survival conditions [56]. Modes of consumption include baked mushrooms, e.g. *Ramaria flava* and *Tricholoma terreum* and which are also used for lichen bread or salad, along with others lihcens such as *Cetraria islandica*, *Evernia prunastri* [58].

Algae biomass is a renewable source of many valuable active substances that have a wide range of applications in many industries, such as food, chemical, agriculture, pharmaceuticals, cosmetics, and medicine [59,60].

Currently, macroalgae are used as novel foods in the food industry, and thanks to their antiviral, anti-inflammatory and anticancer properties, they can be successfully used in therapies [61]. In recent years, there has been an increasing interest in food, cosmetic, and pharmaceutical products of natural origin, which are perceived as healthier and safer for humans [62]. Algae, due to its nutritional value, are presented as “functional foods” or “nutraceuticals” and described as “foods” that contain bioactive compounds, or phytochemicals that may benefit health beyond the role of basic nutrition (e.g., anti-inflammatories, disease prevention). Protein content differs widely across groups of algae and various commercial species of the unicellular green alga *Chlorella* contain up to 70% dry weight protein [59,60]. Among the marine algae, red and green algae such as *Porphyra* spp. (laver), *Pyropia* spp. (nori), *Palmaria palmata* (dulse) and *Ulva* spp. (sea lettuce) often contain high levels of protein in contrast to lower levels in most brown algae. In amino acid composition in marine algae, glutamic acid, and aspartic acid represent the highest proportions of amino acids. These amino acids occur as protein constituents and as free amino acids or their salts. Glutamic acid content decreases after several successive harvests of *Pyronia yezoensis*. Other amino acids (alanine and glycine) also contribute to distinctive flavors of some marine algae [61].

Marine macroalgae do not exceed 2–4.5% dry weight as lipids, mainly as phospholipids and glycolipids [60]. Lipid membranes contain sterols such as fucosterol and beta-sitosterol that have also reported health benefits. Fucosterol occurs in many algae, especially red and brown macroalgae, and it is used in treating the complications of diabetes and hypertension. Algal polysaccharides are the most widely, and often unknowingly, consumed food of algal origin. Edible macroalgae contain high amounts of dietary fiber, ranging from 23.5% (from *Codium reediae*) to 64.0% of dry weight in *Gracilaria* spp. [62].

## 4. Bioactive Compounds of Fungi and Their Health-Promoting Properties

### 4.1. Phenolic Compounds

The growing interest in fungi, including lichen-forming fungi, in recent decades has been associated with the isolation of many bioactive substances in fungi, especially macrofungi, and the documentation of their health-promoting properties [4,58,63,64]. Antioxidants are of particular interest to researchers due to their ability to remove free radicals generated in the human body. Thanks to this, they can counteract many common diseases of civilization, including cancer, atherosclerosis, diabetes, cardiovascular and neurodegenerative diseases. New natural sources of antioxidants are currently being searched for, so that they can be included in the daily diet. The results of many studies indicate that the fruiting bodies of mushrooms, both wild and cultivated, are rich in antioxidants [65,66,67,68,69].

Among the phenolic compounds found in mushrooms, phenolic acids constitute the largest percentage. Due to their strong antioxidant effect and their ability to protect vital structures such as nucleic acids, cell membranes, structural proteins, enzymes and lipids of cell membranes against oxidative damage, they exhibit a broad spectrum of biological activity [70]. The strongest antioxidant abilities (as cell protection against hydrogen peroxide) are demonstrated by vanillic acid, and cinnamic acid derivatives, and caffeic acid [71]. Acids found in fungi such as p-hydroxybenzoic, gallic and protocatechuic acid, are characterized by documented in vitro and in vivo antioxidant, antibacterial, antiviral, antifungal, and anti-inflammatory effects, while also stimulating gastric secretion [72]. Protocatechuic acid also exhibits immunomodulating, spasmolytic, cardioprotective, anticoagulant and chemopreventive effects [73]. Phenolic compounds are the most common antioxidants in the human diet, with their daily supply being about 1 g. The consumption of phenolic acids accounts for about 1/3 of this amount, and these are considered the most valuable antioxidants. Research on the fruiting bodies of edible mushrooms in Poland by Muszyńska et al. [74] showed that the species with the greatest variety of phenolic compounds are *Boletus badius*, *Cantharellus cibarius* and *Pleurotus ostreatus* (four phenolic compounds were found in their fruiting bodies). The presence of protocatechuic acid was demonstrated in all the studied species (1.37–7.50 mg/kg of dry weight); however, the highest amount, 21.38 mg/kg of dry weight, was found in the species *B. badius*. The compound *p*-hydroxybenzoic acid was identified in *B. badius* (1.28 mg/kg of dry weight), *B. edulis* (1.94), and *C. cibarius* (2.30), while its highest content was found in the species *P. ostreatus* (3.60 mg/kg of dry weight). Sinapinic acid, the content of which ranged from 2.11–14.29 mg/kg of dry weight, was also detected in four species: *Armillaria mellea*, *C. cibarius*, *Lactarius deliciosus* and *P. ostreatus*. In turn, cinnamic acid was present in the range of 1.09–4.06 mg/kg of dry weight in *C. cibarius*, *L. deliciosus* and *P. ostreatus*, and its highest content was found in the species *B. badius* (8, 73 mg/kg of dry weight). The compounds *p*-coumaric acid and vanillic acid were found only in single species (respectively: *B. badius* 13.91 mg/kg of dry weight and *C. cibarius* 3.32 mg/kg of dry weight). Conversely, studies by Barros et al. [75,76] in Portugal showed a higher content of phenolic compounds in wild mushrooms compared to cultivated species. Research by Gąsecka et al. [66,67,68] covering 17 species of edible wild mushrooms in Poland showed that species with a high content of phenolic compounds in total were the blewit (*Lepista gilva*) (38.64 mg/g of dry weight) and the brown birch bolete (*Leccinum scabrum*) (22.90 mg/g of dry weight). In these species, the dominant phenolic acids were cinnamic, gallic, vanilla, protocatechin and syringin [77]. Kim et al. [78] determined the content of phenolic compounds in ten species of edible and medicinal mushrooms, in which he showed their higher amounts. Protocatechuic acid was the only one to be found in all the species tested. The same author showed the presence in *P. ostreatus* of gallic acid, homogentisic acid, chlorogenic acid and protocatechuic acid. Differences in the qualitative and quantitative content of phenolic compounds obtained by different authors may result from different genetic properties, or from the different sites from which mushroom fruiting bodies were obtained. They may also result from changing environmental conditions, e.g., air pollution, or the degree of injury to the fruiting body [79]. Other researchers indicate that variation may result from the method of storage and preparation of samples for analysis due to the action of temperature, UV radiation or the phenolic compounds decomposing phenyl oxidase present in mushrooms. Nevertheless, edible mushrooms are an excellent source of valuable antioxidants.

Another example of use for the production of therapeutic agents are lichenized fungi (lichens), due to the properties of their secondary metabolites such as an anticancer, antimicrobial, and antioxidant activity [80,81,82,83,84].

For example, lichen extracts of the genus *Usnea* showed good antioxidant potential, such as a protective role in human lymphocytes by regulating enzymatic activity. This is especially evident in usnic acid (dibenzofuran) isolated from *Usnea* species and other lichens have demonstrated various biological activities, antitumor, antimicrobial and cytoprotective [84]. Additionally, other certain lichen species such as *B. fuscescens*, *P. tiliacea*, and *U. decussata* were potential natural antioxidant resources due to their free radical scavenging activities [85].

Secondary metabolites of lichens and their biological potential use in traditional medicine were considered. Lichens also have the ability to produce phenolic compounds in high concentration, including species of genus such as *Peltigera*, *Solorina*, *Nephroma*, *Cetraria*, *Flavocetraria* and *Alectoria* [86,87]. According to Plaza et al. [88] *Peltigera laciniata*, *Cladonia* aff. *rappi*, *Thamnolia vermicularis* and *Cora* aff. *glabrata* from Venezuelan Andes were natural sources of antioxidants–phenols and could be use in therapy for humans.

### 4.2. Flavonoids

Flavonoid compounds also occur in the fruiting bodies of edible mushrooms. The antioxidant properties of these compounds depend on conjugated double bonds in the C-2 and C-3 position, hydroxyl groups and the carboxyl group in the C-4 position. The mechanism of the antioxidant properties of flavonoids is based on the capture of free oxygen radicals and reactive oxygen species and the limitation of their production in cells by inhibiting the activity of oxidising enzymes (e.g., lipoxygenase), as well as on the easy donation of hydrogen from the carboxyl group, reduction of peroxides and of hydroxides. The indirect mechanism of action is based on chelation of transition metal ions (Cu and Fe), which prevents the formation of reactive hydroxyl radicals in the cells. Moreover, flavonoids can break the cascade of free radical reactions during lipid peroxidation. These compounds have a protective effect on genetic material, cell membranes and enzymes. They also increase the absorption of vitamin C from the gastrointestinal tract, thereby controlling oxidative stress in the body. In the fruiting bodies of chanterelle mushrooms (*Cantharellus cibarius*), the content of flavonoids is 67 mg/100 g of dry weight. Other species in which significant amounts of flavonoids have been shown include the false saffron milk cap (*Lactarius deterrimus*), the cep (*Boletus edulis*), and red cracking bolete (*Boletus chrysenteron*) [29,74,75,89,90,91,92]. Mirończuk−Chodakowska et al. [93] assessed the content of flavonoids in 18 species of edible mushrooms occurring in Poland. The studied species were characterized by a diversified content of flavonoids ranging from 0.84 mg/100 g in the gypsy mushroom (*Rozites caperatus*) to 6.12 mg/100 g in larch bolete (*Suillus grevillei*). A high content of flavonoids was found in species belonging to the genera *Suillus*, *Leccinum* and *Boletus*. In 17 species of edible wild mushrooms in Poland studied by Gąsecka et al. [66,67,68], catechin, vitexin, luteolin, kaempferol, naringenin, apigenin, quercetin and rutin were identified among the flavonoid compounds [77].

For example, according to Plaza et al. [88] the highest level of flavonoids compounds in lichens thalli was detected in ethanolic extract of *Thamnolia vermicularis* from at 37.07 ± 0.08 μg of QE/mg dw extract, while *Peltigera laciniata* showed content only at 9.73 ± 0.07 μg of QE/mg dw extract. The extracts of these lichen species, when tested, were shown to have high anti-oxidant activity.

### 4.3. Carotenoids

Carotenoids are compounds that are synthesized in the organisms of bacteria, fungi, algae and plants. Fungi can synthesize carotenoids containing 4-keto groups of monocyclic structure with thirteen double bonds [94]. These conjugated bonds are responsible for the characteristic color of these compounds (yellow, orange or purple). The role of carotenoids in the human body is to protect cells against the lethal combination of light (photoprotection) and oxidation (antioxidation) and to neutralise singlet oxygen, which can release excess internal energy during the oxidation reaction [95,96]. According to research conducted by the United States of Department of Agriculture, preparations containing carotenoids (e.g., Carotenoid Complex TM), when used systematically, increase the activity of the immune system by 37% after only 20 days, increasing resistance to bacterial and viral infections and to AIDS. The large number of reactive, conjugated double bonds present in carotenoids is responsible for the high activity of these compounds as antioxidants active against free radicals. These compounds stabilise cell membranes and act as photoreceptors. They delay the aging process of the skin, and also prevent irritation resulting from exposure to the sun. The carotenoid complex protects the LDL cholesterol fractions from oxidation, and consequently prevents the formation of atherosclerotic plaques. Consuming more carotenoids reduces the risk of developing cataracts compared to a diet low in these compounds. Edible mushroom species that contain β-carotene include *Cantharellus cibarius*, *Agaricus bisporus*, *Pleurotus ostreatus*, *Boletus edulis*, *Suillus bovinus* (bovine bolete) and *Tricholoma equestre*. Lycopene has been identified in the fruiting bodies of *C. cibarius*, *A. bisporus*, *B. edulis*, *P. ostreatus*, *S. bovinus* and *T. equestre*. Lutein, α-carotene and xanthotoxin have been found in the fruiting bodies of *C. cibarius*, while γ-carotene, auroxanthin and neurosporin have been found in *B. edulis* [29,89,91].

Carotenoids, especially zeaxanthin, are intercellular products (primary metabolites) occurring in lichens, as a typical lichen product from mevalonic pathway [97].

### 4.4. Polysaccharides

Polysaccharides isolated in mushrooms have been shown by numerous studies to have, anticancer, antibiotic, antidiabetic and anti-obesity properties. Therefore, they have great potential in medicine and functional food [98]. One very interesting group of compounds, due to their healing properties, are polysaccharides obtained from fungi, mainly β-glucans. The antitumour and immunostimulatory properties of fungi were first described by [99]. He showed that the administration of boletus mushroom extracts (*Boletus edulis*) to mice suffering from Sarcoma 180 tumours increased their chance of survival. From the fruiting bodies of the giant puffball (*Calvatia gigantea*), Lucas [99] also isolated calvacin, a substance that improves the effectiveness of the treatment of sarcoma and leukaemia. The fruiting bodies of many other species of *Basidiomycota* are also a source of immune-active polysaccharides, as are their mycelial cultures. β−(1 → 3)–and β−(1 → 6) −glucans are a structural element of the interior part of the fungal cell wall. β−glucans linked with chitin chains ensure the simultaneous rigidity and flexibility of the fungal cell wall. These compounds most often have the structure of β-D-glucans or β-mannans [100,101]. Their activity is determined by their molecular weight, solubility in water, number of branches and tertiary structure. The β (1 → 3) glucans with the β (1 → 6) and β (1 → 4) branches have the highest immunoactivity. These polysaccharides in the human body do not cause side effects. However, not all glucans exhibit anticancer properties. This property depends on their water solubility, size and molecular weight, degree of branching, and form of occurrence. High molecular weight β-glucans containing mainly β (1 → 3) bonds are characterized by the highest anticancer properties [102,103]. These polysaccharides in the human body do not cause allergic reactions or adverse effects [8], and show cytotoxic activity against neoplastic cells, as indicated by studies in vitro and in vivo [104,105]. This is especially true for those polysaccharides that in their structure contain other sugars in addition to glucose, such as galactose, xylose, arabinose, mannose and fructose. The antitumour properties of polysaccharides is multivalent and comprehensive. They limit cell DNA damage, reduce the concentration of carcinogens in the human body and inhibit their activation, as well as inhibiting the development of cancer cells. Thanks to their ability to activate the immune system, they can support the treatment of not only cancer, but also infectious diseases. An important advantage of naturally derived polysaccharides is that they buffer the side effects of chemotherapy and, moreover, do not exhibit any toxic effects on the human body [8,106]. Anticancer drugs derived from fungi, namely polysaccharides, have been used in the treatment of cancers of the gastrointestinal tract, breast, cervix and lungs. Examples of such drugs of fungal origin are the following preparations isolated from fruiting bodies, mycelium or the culture medium of *Basidiomycota* mushrooms: Schizophyllan isolated from the culture medium *Schizophyllum commune*, Grifolan isolated from the fruiting bodies and mycelium of *Grifola frondosa*, Krestin isolated from the mycelium of *Trametes versicolor*, and Lentinan isolated from the fruiting bodies of *Lentinula edodes*. These have been introduced to official therapy regimes by, among others, Japanese researchers [8,107]. The use of *T. versicolor* extracts also acts on the induction of superoxide dismutase (SOD), which inhibits the multiplication of cancer cells [108].

A less well-known group of polysaccharides is α−(1 → 3)−glucans. They occur in the innermost layer of the fungal cell wall, fulfilling a structural and supporting function, and constitute a reserve material. It was found that α-glucans exhibit anticancer, immunostimulating and antioxidant properties [109]. The biological activity of α−(1 → 3)−glucans depends on their solubility, structure and concentration. The activity of insoluble α-glucans can be increased by chemical modification. It was shown that derivatives obtained by carboxymethylation of α−(1 → 3)−glucans were characterized by high cytotoxic activity [110]. Many polysaccharides have been isolated from wild fungi [111]. These have been shown to be characterized by a wide range of biological activity, mainly anticancer, anti-inflammatory and antioxidant [105,112,113]. One of the species that has been widely studied for many years is the lingzhi mushroom (*Ganoderma lucidum)*. It has been known in the folk medicine of Asian countries for about 4.000 years [114]. The use of *G. lucidum* has recently become popular in Europe, as well as for their nutritional and health benefits, especially its anticancer functions [115].

*Ganoderma* *lucidum* is used in the treatment of liver diseases, hypertonia, arthritis, asthma and stomach ulcers. For several decades, this mushroom has been obtained from breeding, which has made it more widely available. In China, this mushroom is known as lingzhi, and in Japan, as reishi [116]. Another species that has been studied in recent years is the shiitake (*Lentinula edodes*). It has been shown that preparations from mycelial extracts of LEM (*Lentinus edodes* mycelia) and those obtained from an aqueous LEM solution by precipitation with ethyl alcohol, LAP, have very strong anticancer properties involving the activation of the immune system [5]. Apart from polysaccharides, these preparations also contain nucleic acid derivatives, ergosterol and B vitamins, especially B1 (thiamine), B2 (riboflavin). The anticancer properties of polysaccharides are enhanced by the presence of protein [86]. Lignins obtained from the culture of *Lentinula edodes* are being investigated as potential drugs in the treatment of hepatitis B and AIDS. The polysaccharides produced by shiitake (*Lentinula edodes*) are also characterized by immunomodulatory effects and are used in chemoprevention and as an adjunct in cancer therapy, also alleviating the undesirable effects of chemotherapy [98]. A species that has been widely studied in recent years is also the lion’s mane (*Hericium erinaceus*). The bioactive substances extracted from it exhibit anticancer properties and may also be important in alleviating the symptoms and effects of Parkinson’s and Alzheimer’s diseases [117]. They also strengthen the body, improve digestion, and are helpful in the treatment of certain cancers [118]. The lion’s mane mushroom produces several types of biologically active substances, including intracellular polysaccharides (IPS) and extracellular polysaccharides (EPS), which exhibit immunostimulatory and anticancer properties [102,103]. Previous studies, as reported by [119], have shown that the compounds present in *Hericium erinaceus* have many health-promoting properties. Polysaccharides isolated from this mushroom showed antitumor activity, among others, cancer of the stomach and intestines. *Hericium erinaceus* is a fungus with brain-enhancing properties. It also causes changes in both the composition and activity of the gastrointestinal microflora, resulting in nutritional and health benefits [120]. Hericenones and erinacines compounds present in the lion’s mane mushroom affect the development of cells in the brain. Hericenones stimulate the activity of nerve growth factor (NGF) synthesis. It is assumed that deficiency of NGF is related to Alzheimer’s diseases. Bioactive compounds isolated from *Hericium erinaceus* have been applied to the treatment of Alzheimer’s disease [121]. Studies with myoma-loaded mice have shown that the lion’s mane mushroom has anti-tumor effects. The ethanolic extract of this fungus strongly inhibited collagen-induced platelet aggregation [122]. Due to the protected status of this species in many countries, including Poland, where this study was conducted, a viable source of immunoactive polysaccharides in the case of the lion’s mane may be provided both by the fruiting bodies and their mycelial cultures [123]. Lectins are compounds with multidirectional, pro-health effects occurring in mushroom fruiting bodies [105]. Lectins include polysaccharide-protein or polysaccharide-peptide complexes. They display anticancer, immunomodulatory and antiviral properties [124,125]. Lectins isolated from different species of fungi differ in molecular weight, number of subunits and type of carbohydrate bound [126]. Lectins with different biochemical properties have been isolated from the same species [127]. Lectins have also been identified in different parts of the fruiting body, i.e., in the cap and body, and in the mycelium. Their activity is also strictly dependent on the age of the fruiting body and the season [128]. A lectin with a particular specificity for xylose and melibiosis was isolated from *B. edulis*. It showed potential mitogenic properties and an inhibiting effect on the HIV−1 enzyme, reverse transcriptase, which may indicate the possibility of using lectin in the prevention of immunosuppression in patients after chemotherapy and radiotherapy or in AIDS patients [129]. Despite the large variety of mushroom lectins, little information exists about their function and biological activity.

Currently, intensive research is carried out on the chemical nature and mechanisms of action of fungal metabolites. These substances found in medicinal mushrooms such as *Fomitopsis pinicola*, *Inonotus obliquus*, *Hericium erinaceus*, *Trametes versicolor* and their therapeutic potential against cancer are described by [130]. The anti-tumor properties can be realized not only by the inhibition of certain cancer-specific processes or the targeted activation of tumor-specific apoptosis, but also through actions such as immunomodulation. In particular, *Trametes versicolor* are useful as pharmaceutical due to their strong and complex immunomodulatory potential provided by rich polysaccharide and proteoglycan diversity [130]. Many researchers have reported the presence of immunostimulatory polysaccharides such as tham lichenan, isolichenan, and β-glucan in lichens. Anticancer and antiviral (HIV) properties are also demonstrated by lichen extracts containing their metabolites. The literature data presented of secondary metabolites such as: polysaccharides type of *β* (1 → 6) glucan with acetyl group isolated from *Umbilicaria esculenta*. This compound, as noted, could significantly inhibit the expression of human immunodeficiency virus (HIV) antigen on Molt-4 cells. The study also reported antiviral activity of a polysaccharide isolated from the lichen thallus *Parmelia perlata* against yellow fever virus [131]. In addition, researchers have shown in their study that polysaccharides extracted from *Lobaria pulmonaria* could be a potential drug for Alzheimer’s disease [132]. As shown in the literature review (+)-protolichesterinic acid from *Cetraria islandica* cold inhibit the cell vitality of human breast cancer [4]. The extract from *Usnea longissima* containing barbituric acid could effectively inhibit the cancer cells of the cervix, lung, breast and prostate. Norstictic acid extracted from *Ramalina* sp. had cytotoxic activity against UACC-62 human melanoma cells. There are many studies which describe the anticancer activity of lichens. Lichen’s metabolites exert their antitumor activity through their cytotoxic activity, cell cycle regulation, anti-proliferation, anti-invasiveness, anti-migration, anti-angiogenesis, telomerase inhibitory, inhibition of endothelial tube formation, and other pathways [133]. Secondary metabolites isolated from lichens thalli have an effect on cells of lung, prostate, breast, colon, liver, cervical, rectal, pancreatic, ovarian and lymphatic cancer as well as melanoma, cancer, cancer, leukemia, glioblastoma, astrocytoma and other cell lines. The potential health benefits of lichens for antioxidant and anticancer activities are still unclear and need to be further studied [4].

### 4.5. Lndole Compounds

Edible mushroom species are a source of non-hallucinogenic indole compounds and their derivatives. This group of compounds includes L-tryptophan, 5-hydroxytryptophan, serotonin, melatonin and tryptamine [134,135,136]. Indole compounds are neurotransmitters or their precursors, which also have anticancer and anti-aging effects, and regulate the daily cycle of the human body and participate in the process of blood coagulation. Antioxidant properties are also associated with the presence of indole compounds in mushrooms. These compounds are very important due to their potential role in counteracting depression and neurodegenerative diseases, e.g., Parkinson’s and Alzheimer’s. One example of mushrooms used in traditional medicine is *H. erinaceus*. This mushroom has great potential in the treatment of neurological disorders due to its high content of neuroactive compounds because it is a rich source of indole compounds (5-hydroxy-L-tryptophan (5-HTP), melatonin and tryptamine) and has a high phenolic content [137,138]. These compounds and their derivatives can be used as anti-inflammatory and analgesic drugs. Studies of methanol extracts from the fruiting bodies of edible mushroom species popular in Poland such as *Agaricus bisporus* (common mushroom), *Boletus edulis* (cep mushroom), *Boletus badius* (brown boletus), *Cantharellus cibarius* (chanterelle), *Lactarius deliciosa* (saffron milkcap), *Leccinum rufum* (orange bolete), *Pleurotus ostreatus* (oyster mushroom) and *Suillus luteus* (slippery jack) as well as conditionally edible mushrooms such as *Armillaria mellea* (honey fungus), *Lactarius deterrimus* (false saffron milk cap), and *Tricholoma equestre* (yellow knight) conducted by [74] showed the different content of indole compounds in individual species. Serotonin, which belongs to this group of compounds, is a natural neurotransmitter regulating the human daily cycle. It is widely researched and used as a mood enhancer in a daily dose of 100 to 200 mg, and in the treatment of depression and migraines up to 600 mg. This compound is also a potential drug in the treatment of Alzheimer’s disease [139]. Serotonin also has antioxidant properties due to its role in the reduction of lipid peroxidation. The high content of serotonin in the above-mentioned edible species, which are among the most popular in Europe and Asia, means that they can be considered a good food source, rich in serotonin, additionally confirming the dietary and therapeutic value of edible mushrooms [140]. In conditionally edible species, a low serotonin content was found (maximum 2.20 mg/100 g of dry weight in *A. mellea* and 0.18 in *T. equestre*), though a higher tryptamine and tryptophan content (from 2.01 to 4.46 mg/100 g of dry weight) was noted than in the previously mentioned typically edible species. Tryptamine may interact with drugs from the group of MAO inhibitors, and in people taking these derivatives, this may lead to fatal poisoning [141]. Tryptophan administered in excessively high doses damages the nervous system and may contribute to the induction of bladder tumours.

Surprisingly, secondary metabolites of hallucinogenic mushrooms with psychoactive properties can be used as drugs. The most famous hallucinogenic substance present in wild mushrooms is psilocybin (3-[2-(Dimethylamino)ethyl]-1*H*-indol-4-yl dihydrogen phosphate). It has been found in fruiting bodies of lamellar fungi, which are part of the *Psilocybe* genus. Psilocybin-assisted therapy in producing large, rapid, and sustained antidepressant effects among patients with major depressive disorder was successful [142].

Not only mushrooms have the ability to synthesize hallucinogenic compounds such as psilocybin. It is remarkable to find the presence of indole compounds–tryptamine and psilocybin also in the lichen thallus *Dictyonema huaorani* occurring in Ecuador. The analysis showed the likely presence of 5-methoxy-N-methyl-tryptamine, 5-methoxy-dimethyl-tryptamine, and 5-methoxy tryptamine [143].

### 4.6. Sterols and Vitamin

The first reports of the presence of sterols in mushroom fruiting bodies date back to 1887. The sterols found in fungi are distinguished by a high degree of unsaturation and are called mycosterols [144]. Mushrooms are one of the few sources of ergosterol, a precursor to vitamin D2. The two main physiological forms of active vitamin D are ergocalciferol (D2) and cholecalciferol (D3). Ergosterol can be converted to vitamin D2 through the action of UV radiation. Ergosterol is photolyzed by UV radiation with a wavelength of 280–320 nm, giving various products. The main ones are provitamin D2 and the ergosterol isomers, tachysterol and lumisterol. Provitamin D2 undergoes spontaneous rearrangement into vitamin D2 under the influence of heat. After ingestion of vitamin D2, it is metabolized to the biologically active form 1α, 25-dihydroxyvitamin D2 through the intermediate form 25-hydroxyvitamin D2. A similar process takes place in human skin, especially in the epidermis (mainly in the keratinocytes of the reproductive layer). The 7-dehydrocholesterol, e.g. provitamin D3, undergoes non-enzymatic photo-isomerisation to provitamin D, and then within a few hours, under the influence of thermal energy, it is converted to vitamin D3. In the plant and animal kingdom, ergosterol and vitamin D2 are virtually absent, only being present in a few species of fish, such as salmon, cod, tuna (and their fish oil). Therefore, the main source of vitamin D, in both northern and southern latitudes, are mushrooms. Vitamin D makes calcium more digestible for children, the elderly and women after menopause. Vitamin D deficiency can lead to osteoporosis in children and in adults. Studies have also shown significant effects of vitamin D in reducing the risk of breast cancer, colorectal cancer, prostate cancer, autoimmune diseases, especially type I diabetes, Crohn’s disease and cardiovascular diseases [145]. *Cantharellus cibarius*, or chanterelle, is one of the most valued and most often collected species of mushrooms. It has a high nutritional value, an aromatic scent and an eye-pleasing orange color and is also not susceptible to insect infestation. Chemical composition studies of this species have shown that it is a rich source of ergocalciferol. In addition, even after several years of storage of the dried fruiting bodies, the vitamin D2 content remains high, on average 1.43 μg/g of dry weight. The differences in the ergocalciferol content result from the different sun exposure of the sites from which the fruiting bodies are derived [146]. The fruiting bodies of *Boletus edulis*, the cep mushroom, are also a rich source of microsterols. On the basis of chemical analysis, it has been shown that the entire fruiting body contains approx. 200 mg of vitamin D2 per 100 g of dry weight. The fruiting bodies of *B. edulis* contain approximately 500 mg of ergosterol/100 g of dry weight. Moreover, ergosterol peroxide (30 mg/100 g of dry weight) was isolated. Ergosterol peroxide is a steroid with a broad spectrum of biological properties, including antibacterial and anti-inflammatory properties, as well as being toxic to various tumour cell lines. Three other mycosterols related to ergosterol have been isolated from boletus: ergosta-7-enol (16.4 mg/100 g of dry weight), ergosta-5,7-dieneol (12.5 mg/100 g of dry weight), and ergosta-7,22-dienol (11.2 mg/100 g of dry weight). These compounds are characterized by strong antioxidant and anticancer properties. The relatively high content of the above compounds in the fruiting bodies of *B. edulis* makes them a good source for vegetarians and vegans. *Lentinula edodes,* the shiitake mushroom, is also characterized by a high content of microsterols. Nowadays, it is used to treat what are commonly known as the diseases of civilization. The ergosterol content in shiitake is on average 85 mg/100 g of dry weight. In addition, three other mycosterols are present: ergosta-7.22-dienol, ergosta-5,7-dienol and ergosta-7-enol [144]. Apart from vitamin D precursors, the presence of other vitamins in the fruiting bodies of wild mushrooms, such as tocopherol (vitamin E), β-carotene (provitamin A), and ascorbic acid (vitamin C), has also been noted. These compounds, apart from polyphenols, determine the antioxidant properties of fungi [147,148]. One species with a high vitamin content is *Cantharellus cibarius* [149,150]. The content of total tocopherols in the fruiting bodies of this species is 4.33 mg/g (α−tocopherol–1.25 mg/g, β−tocopherol–1.79 mg/g, γ−tocopherol–1.29 mg/g). The content of β-carotene was 0.79 mg/100 g, and of ascorbic acid was 0.99 mg/100 g. The presence of lycopene, a carotenoid pigment (0.33 mg/100 g), was also noted in the fruiting bodies [151].

An interesting example of a mushroom with a broad spectrum of medicinal properties is *Ophiocordyceps sinensis*. It contains many other compounds classified as sterols, e.g., H1-A. It is similar to ergosterol and has been shown to have no glucocorticosteroid receptor binding ability. The ability to bind glucocorticosteroid receptors has been considered. H1-A is a type of ergosterol, and its structure looks like testosterone and dehydroepiandrosterone. Many of the compounds synthesized by *Ophiocordyceps sinensis* also have anticancer activity, and much more [152].

Sterols have been isolated and characterized from various lichen species, for example, *Xanthoria parietina*, *Pseudevernia furfuracea*, *Lobaria pulmonaria* and *Ramalina africana*. Brassicasterol, lichesterol, poriferasterol and B-sitosterol have been isolated from lichens thalli [26].

## 5. Algae Bioactive Compounds and Their Health-Promoting Properties

Due to their antibacterial, antiviral, antifungal and anti-inflammatory properties, algae can be used to treat many diseases [153,154]. The healing properties of algae have been known and exploited for many years [155,156]. They have been successfully used in respiratory diseases, e.g., bronchitis, colds, and chronic cough, against helminthiasis, and for slimming [156], against venereal disease, enlarged thyroid gland, and gout, as well as in ointments and anaesthetic [62]. These properties result from the biologically active compounds present in the algae (Table 2).

The group of proteins and amino acids in algae includes glycoproteins and metalloproteins, as well as exogenous amino acids (alanine, asparagine, glycine, lysine, serine, isoleucine, leucine, methionine, phenylalanine, threonine, tryptophan and valine) [151,154]. Among the lipids in algae there are essential unsaturated fatty acids (EFAs) including arachidonic acid, eicosapentaenoic acid and the rare γ-linolenic acid (GLA) [154,159,160]. In the group of vitamins in algae, carotenoids have been identified, e.g., β-cartene, a source of vitamin A, vitamins from the B group (B1, B2, B5, B6, B12) and vitamins E (tocopherol), C (ascorbic acid) and D [154,159,160]. The micronutrients present in algae include Br, Zn, I, Mg, Mn, Cu and Fe. They occur in a particularly easily digestible form as organometallic or complex compounds [161,162]. Other chemical compounds identified in algae include polyphenols [154,163] and natural plant pigments such as phycoerythrin, phycocyanin and chlorophyll [154,159,160,162,163].

Microalgal genera commonly considered as beneficial dietary supplements include *Chlorella*, *Odontella*, *Porphyridium* and *Scenedesmus*, with species *Chlorella* being recognized as particularly rich in polysaccharides [164]. This bioactivity of algae contains anticancer properties, cytokine modulation, ani-inflammatory effects, macrophage activation, and inhibition of protein tyrosine phosphatase. Acidic polysaccharide extracts from *Chlorella pyrenoidosa* have been patented as potentially useful antitumor and immunostimulating supplements [59,61]. Alginate is the major polysaccharide of brown algae, comprising 14–40% of its dry mass and was isolated as algin from kelp (*Laminaria* sp.). The abundant, heavily sulphated ulvans are extracted from members of the Ulvales. Bioactivities of ulvan extracts in vitro include antibacterial, anticoagulant, antioxidant, antiviral, antitumor and antihyperlipidemic effects. The main polysaccharides after the alginates in brown algae are beta-glucans (laminarans), cellulose, and heteroglycans [164].

### 5.1. Carotenoids

Many compounds obtained from algae, such as carotenoids, flavonoids and phenolic acids, exhibit anticancer properties [165,166,167]. Ishikawa et al. [168] demonstrated the anticancer properties of carotenoids isolated from brown algae *Undaria pinnatifida*. Fucoxanthin and its deacetylated metabolite (fucoxanthinol) have been shown to inhibit the viability of HTLV-1 infected T cell lines (*Human T-cell leukemia virus type 1*). These carotenoids showed more potent inhibitory effects than astaxanthin or *β*-carotene. The human T-lymphocytotropic retrovirus HTLV-1 is responsible for the development of adult T-cell leukemia–ATL (*adult T-cell leukemia*). The cells most frequently attacked by HTLV-1 are CD4 + -0669 T cells. Studies have shown that fucoxanthin and fucoxanthinol can be a potential therapeutic preparation used in patients suffering from ATL. The relationship between the I content of algae and the risk of breast cancer was investigated by [169]. An aqueous solution of powdered brown algae *Undaria pinnatifida* showed a strong inhibitory effect on the development of carcinogenesis in rat mammary glands, while no toxic effect was reported. Under in vitro conditions, the solution of *Undaria pinnatifida* strongly induced apoptosis of the three types of human breast cancer cells. This effect was much stronger than that achieved with 5-fluorouracil (5-FU), a widely used anticancer drug, 1678-0689 in women with breast cancer. In Japan, these algae are a cheap and safe food source. Research suggests that brown algae *Undaria pinnatifida* could be used in the chemoprevention of breast cancer in women. Algae produce in addition to fucoxanthin other carotenoids such as astaxanthin, lutein and zeaxanthin. Canthaxanthin is synthesized by *Chlorella vulgaris*, *Haematococcus pluvialis* and shows antioxidant, anti-cancer, anti-diabetic activity. It also, prevents obesity [157].

### 5.2. Polysaccharides

Polysaccharides constitute approx. 60% of all active substances present in algae. This group of chemical compounds includes mucopolysaccharides (GAG glycosaminoglucans), compounds made of amino sugars and uronic acids. The most well-known glycosaminoglycans include hyaluronic acid and chondroitin sulphate, alginic acid and its salts (especially calcium and sodium alginates), as well as fucans (laminarin and fucoidin), mannitol and sorbitol (polyalcohols, so-called sugar alcohols, combination of carbohydrates with alcohol), carrageenans (natural linear sulphated polysaccharide) and agar (a natural gelling and thickening agent) [154,155,156,161,170]. Brown algae also contain large amounts of alginic acid and fucoidan, a polysaccharide, which is a typical compound for this group [157]. Fucoidan is a polysaccharide composed of α-L-fucopyranose molecules that can only be linked together by 1 → 3 bonds or 1 → 3 and 1 → 4 alternating bonds. The branched structure of fucoidan is due to the attachment to the main chain of α-L-fucopyranose residues, as well as inorganic and organic substituents such as sulfate (VI) residues, D-glucuronic acid and acetyl residues. Fucoidan also includes monosaccharides, such as galactose, xylose, mannose, glucose and uronic acid, but their location in the polymer remains unknown [171,172,173]. Fucoidan, extracted from brown algae *Eclonia cava*, *Sargassum hornery* and *Costaria costata*, found on the Korean coast, can be used in cancer prevention by inhibiting the growth of cancer cells (melanoma and colon cancer) [167]. Fucoidan is used in pharmaceutical and cosmetic preparations as a compound extending shelf life. Fucoidan is obtained from the intercellular substance of brown algae, such as *Fucus vesiculosus* (bladder wrack), *Fucus evanescens*, *Fucus serratus* (toothed wrack), *Fucus distichus*, *Laminaria saccharina*, and others [161]. Fucoidans obtained from various types of brown algae differ in chemical structure, strength and activity profile [161,174]. Two types of fucoidan are currently described: F-fucoidan, which is 95% sulfonated glucose esters, and U-fucoidan, which is composed of 20% glucuronic acid. On the basis of tests conducted on an animal model in vitro and in vivo, no toxic effects of fucoidan from various brown algae were found, even at very high levels of consumption. Also, on the basis of clinical trials, no negative effects of fucoidan on the human body were noted [171,175]. The conducted studies confirmed the anticancer, anticoagulant, antiviral, anti-inflammatory and antioxidant effects of fucoidan.

Another polysaccharide with important medicinal and nutritional significance for humans is agar. Agar is a water-soluble long-chain polysaccharide composed of agarose and agaropectin. The highest amounts of it are found in red algae. The best sources of agar are algae from genus *Gracilaria*, *Gigartina* and *Gelidium*. This substance is widely used in the production of gelatin. As studies have revealed, agaro-oligosaccharides have the ability to inhibit the production of a pro-inflammatory cytokine and an enzyme associated with nitric oxide production. It exhibits blood glucose lowering, anti-aggregation, antioxidant, anticancer and photoprotective effects [176,177].

A polysaccharide of great importance is laminarin. This compound is a water-soluble polysaccharide composed of (1,3)-β-D-glucan with β-branching (1,6) and containing 20–25 glucose units. It is produced by algae such as *Saccharina latissimi*, *Fucus vesiculosus*, *Ascophyllum nodosum*, *Undaria pinnatifida* and numerous species of the genus *Laminaria*. It was shown that it influences intestinal pH level, mucus structure and production of short-chain fatty acids. Laminarin has anticancer, antibacterial and photoprotective properties. Additionally, and very important, it has an immunostimulating effect, lowers blood pressure and reduces the level of cholesterol, triglycerides and phospholipids in liver [25,178].

Another compound is alginate. Its properties with medical potential include chelating metals, reducing cholesterol and blood pressure. It also has detoxification abilities. It is synthesized by brown algae [25].

Ulvan produced mainly by algae of the genus *Ulva* shows anticoagulant activity. It is used to treat stomach ulcers and flu. Very valuable are compounds produced by Red algae belonging to *Porphyridium* but also interestingly by cyanobacteria–*Nostac flegelliforme*. It is porphyran showing antioxidant and anticoagulant activity. Importantly, it also exhibits antiviral activity against Herpes simplex virus (HSV-1 and HSV-2). It stimulates the immune system and lowers blood lipid levels [179].

Sulfate polysaccharides (containing hemiester sulfate groups in their sugar residues) exhibit immunomodulatory, antitumor, antithrombotic, anticoagulant, antimutagenic, anti-inflammatory, antimicrobial, and antiviral activities including anti-HIV infection, herpes, and hepatitis viruses [170]. Red algae (*Chondrus crispus*, *Porphyra tenera*, *Schizymenia binderi*) produce sulfated galactans consistin of the β-galactose or α-galactose units, which have anticoagulant activities. Mišurcová et al. [170] reported that 2,3-di-o-sulfated D-galactan from *Botryocladia occidentalis* exhibited anticoagulant activity comparable to heparin.

### 5.3. Sterols and Vitamin

Algae are rich source of phytosterols and important components of healthy diets. Phytosterols reduce blood cholesterol levels in hyper and normocholesterolemic subjects and may also inhibit colon cancer development [180]. They are structurally similar and functionally analogous to cholesterol in vertebrate animals. Milovanovic et al. [181] suggested that they are essential constituents of cellular membranes, playing important functions in the control of membrane fluidity and permeability, and signal transduction as hormones or hormonal precursors. Algae possess some specific phytosterols such as fucosterol (found in Phaeophyceae) and desmosterol (which is a biosynthetic precursor of fucosterol). Brown algae mainly contain C_27_, C_28_ and C_29_ sterols. Kapetanovic et al. [182] studied the sterol composition of marine algae from the Adriatic Sea and found in *Ulva lactuca* cholesterol and isofucosterol, and in *Cystoseira adriatica* were cholesterol stigmast-5-en-3-ol. Fucosterol was described as the predominant sterol in brown algae (83–97% of total sterol content), whereas desmosterol was the main compound in red ones (87–93% of total sterol content). Lopeset et al. [183] described the phytosterol profiles of 18 macroalgae from the Portuguese coast. C_29_ sterols were the main compounds in brown and green alga species (71–95 of total sterol content). *Cystoseira tamariscifolia* was the species with the highest phytosterol content and in this brown algae, seven sterols were identified, namely desmosterol, ergosterol, fucosterol, cholesterol, sitosterol, campesterol, and stigmasterol. Concerning red algae species, cholesterol was generally the major compound, *Osmundea pinnatifida* having the highest cholesterol and total phytosterol content. Li et al. [184] found fucosterol, ergosterol, and cholesterol in *Ecklonia cava*, while Yoon et al. [185] identified 24-hydroperoxy-24-vinylcholesterol in *Eclonia stolonifera* for the first time [186,187].

### 5.4. Polyphenols

Polyphenols are a heterogeneous group of compounds containing many biopolymers with unique structure and biological properties found in seaweed at high concentrations [157,188]. Algae (green, brown and red algae) accumulate phenolic compounds such as bromophenols, phenolic acids, flavonoids, phloroglucinol and its polymers such as phlorotannins [188,189]. Terrestrial plants accumulate only 2–3% of polyphenols, while levels in *Fucus* reach 3–12% of dry weight and in *Ascophyllum nodosum* 14% [188]. The largest amount of phlorotannins, accumulated within the vegetative cells of the outer cortical layer of the thalli, in Japanese Laminariaceae (*Eisenia bicyclis*, *Ecklonia cava*, *Ecklonia kurome*) was investigated [156,157]. Besednova et al. [188] noted that concentration of polyphenols in algae was significant depending on the species, distributions, tissue type, water salinity, amount of nutrients, and extraction methods. Extracts from seaweeds containing polyphenolic metabolites are bioavailable, non-toxic to eukaryotic cells and, at the same time, have a pronounced bactericidal or bacteriostatic effect on a wide range of pathogenic microorganisms [188]. Polyphenols due to variety of biopolymer types and multifunctional properties (antioxidant, antiviral, antithrombotic, fungicidal, neuroprotective and antitumor) demonstrate the high potential for the medicinal application.

## 6. Conclusions

Medicinal mushrooms, lichens and algae are a source of a large number of biologically active substances, many of which are currently used in medical therapies. In recent years, polysaccharides with immunoregulatory and anticancer properties contained in some fungi and algae have been intensively researched. Thanks to their ability to activate the immune system, these can support the treatment of cancer. An additional advantage of naturally derived polysaccharide compounds is the fact that they often buffer the side effects of chemotherapy, and that they do not exhibit toxic effects on the human body. Fungi and algae can also be consumed directly and can be treated as a type of functional food. The positive effects of fungi and algae result from the interaction of various active ingredients contained in their organisms. It is also possible to use ready-made preparations of fungal origin in the form of capsules or tablets containing purified fungal or algae extracts, treated as dietary supplements. They are most often used as preparations to strengthen the immune system and prevent cancer. Preparations of this type are mainly produced from such species of medicinal fungi as *Ganoderma lucidum* (lacquered bracket or reishi), *Lentinula edodes* (shiitake), *Grifola frondosa* (hen-of-the-woods), *Trametes versicolor* (turkeytail), *Ophiocordyceps sinensis* (caterpillar fungus), and in the case of algae from genera such as *Chlorella* and *Spirulina*. There are also preparations containing extracts from various species of fungi, including lichens species such as *Cetraria islandica*, *Umbilicaria esculenta* and species from genera *Cladonia*, *Usnea* and many others.

The natural organisms, such as fungi, lichenized fungi and algae, could be used as supplementary medicine, in the form of pharmaceutical preparations and food sources. Further advanced studies are required on the pharmacological properties and bioactive compounds of these organisms. The impact of fungi and algae on human health can be beneficial, opening up new perspectives for studying their biological properties. Due to bioactive properties of these organisms, it may contribute to health benefits for consumers.

## Figures and Tables

**Table 1 nutrients-13-03178-t001:** Biologically active compounds extracted from fungi from Poland in g/100 g of edible parts [8,37,38].

Species	Water	Protein	Lipids	Polysaccharides	Cellulose
*Agaricus bisporus*	90.9	2.4–3.1	0.4–0.5	4.8–5.5	3.4
*Boletus edulis*	87.2	2.8–3.6	0.4–0.5	5.8–6.0	3.8
*Cantharellus cibarius*	89.1	1.5–1.6	0.5–0.8	4.7–6.6	3.2
*Lactarius deliciosus*	82.6	1.9–2.3	0.7–0.9	6.9–7.2	–
*Leccinum rufom*	86.6	1.5–2.0	0.8–0.9	4.7–5.0	3.9
*Leccinum scabrum*	–	3.1–3.4	0.6–0.8	6.5–6.8	–
*Suillus luteus*	90.8	1.7–2.1	0.4–0.9	5.1–5.9	2.7
*Xerocomus subtomentosus*	90.4	2.0–2.4	0.2–0.5	3.8–4.5	2.4

**Table 2 nutrients-13-03178-t002:** Biologically active compounds extracted from micro- and macroalgae (seaweeds).

**Compounds**	**Microalgae**	**Macroalgae References**
Proteins	*Ulvales*	glutamic acid (*Pyropia yezoensis*) [61] *Rhodophyta* [20,27]
Lipids	DHA EPA (*Trachydiscus minutus*, *Odontella aurita*, *Phaeodactylum tricornutum*)	fatty acids (*Rhodophyta*, *Phaeophyta*) [22,153]
Minerals	aquamin	*Rhodophyta* [150]
Vitamins	Tocopherol riboflavin (*Chlorella stigmatophora*) cobalamin (*Dunaliella tertiolecta*) thiamine (*Volvox carteri*) ascorbic acid	*Macrocystis pyrifera* [153] *Pyropia yezoensis* [21,153,156] [153] *Porphyra umblicalis*
Polysaccharides	ulvans (*Ulvales, Chlorophyta*)	alginate (*Laminaria* sp.) [21,157] agar (*Gracilaria* cornea) [157] fucoidans (*Phaeophyta*) [153,158] laminarans (*Phaeophyta*) [61] carrageenans (*Gelidiales, Gigartinales, Gracilariales*) [155,156] porphyran (*Rhodophyta*) [156,157]
Antioxidants	astaxanthin (*Haematococcus pluialis*, *Chlorella zofingiensis*)	ascorbate [55,59] glutathione [151]
Bioflavonoids	rutin quercetin kaempherol	[20,154]
Polyphenols	carboxylic acids (*Bacillariophyceae*, *Eustigmatophyceae*, *Chlorophyta*) hydroxycinnamic acids (*Chlorella vulgaris*, *Haematococcus pluvialis*)	phlorotannins (*Phaeophyta*) [60,154] catechins [156] flavonoids [153,155] tannins [156] lignans [156] mycosporine [151] bromphenols (*Rhodophyta*) [156]
Phlorotannins		*Phaeophyta* *Ascophyllum nodosum* [60]
Pigments	chlorophyll b (green algae)	fucoxanthin (*Phaeophyta*) [62,156] peridinin (*Dinoflagellate*) phycoerythrin (*Rhodophyta*) phycocyanin (*Rhodophyta*) [153,156]
Unsaturated fatty acids	green alga (*Ulva lactuca*)	[59,155]
Sterols	diatoms (*Gomphonema*), green algae *(Ankistrodesmus*, *Monoraphidium, Scenedesmus*)	brassicasterol (*Palmaria decipiens*) fucosterol (*Rhodophyta*, *Phaeophyta*) [61,155,156]
Phytohormones	auxin, abscisic acid, cytokinin (*Chlorophyta*)	[59,154,155]

## Data Availability

The data presented in this study are available on request from the corresponding author.

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
