# Peer review of "Fungi and Algae as Sources of Medicinal and Other Biologically Active Compounds: A Review"

_nutrients, 2021, doi:10.3390/nu13093178_

Round 1
Reviewer 1 Report
Comments for Nutrients-1368848
The study aims to characterise the bioactive substances and nutritional value of edible fungi mushrooms, including lichenized fungi (lichens) and algae. Especially focus on the bioactive component on human health-promoting properties of individual species. They summarized the natural compounds classified according to their occurrence, structure and biological functions in Table 1, and the biologically active compounds extracted from micro- and macroalgae in Table 3.
However, the main biologically active compounds extracted of fungi mushrooms, such as polysaccharides and polyphenols, may have been well investigated already. For example:
Besednova et al. Antiviral Effects of Polyphenols from Marine Algae. 2021;9(2):200.
Mišurcová et al. Health benefits of algal polysaccharides in human nutrition. Adv Food Nutr Res. 2012;66:75-145.
The authors also concluded by themselves “In recent years, polysaccharides with immunoregulatory and anticancer properties contained in some fungi and algae have been intensively researched” in their conclusion section.
Furthermore, some review papers had summarized the functions of bioactive components in fungi mushrooms. Such as Shahidi &Rahman reported the bioactives in seaweeds, algae, and fungi and their role in health promotion (J. Food Bioact. 2018;2:58-81).
Taken together, I believe it is really a hard work to summarize this review paper. But the beneficial of fungi and algae on human health has been recognized for a long time, may be not a new issue for the readers.
Author Response
Answers to points raised by the Reviewers of manuscript ID: nutrients-1368848
"Fungi and Algae as Sources of Medicinal and Other Biologically Active Compounds: A Review” by Joanna Åšlusarczyk et al.
I would like to thank you very much for the remarks about our manuscript titled “Fungi and Algae as Sources of Medicinal and Other Biologically Active Compounds: A Review”.
I am grateful to Reviewers for their comments. These comments allowed me to improve the quality of the manuscript. Below, I listed my answers to all questions and points raised by the Reviewers. All changes in revised manuscript are marked in red font.
Answers to the Reviewer’s comments:
Comment |
Answer |
Reviewer 1 |
|
However, the main biologically active compounds extracted of fungi mushrooms, such as polysaccharides and polyphenols, may have been well investigated already. |
Thank you for this comment. In Introduction section I explained why this manuscript is important and presents a new approach to the issue. I also added new references and revised all references. I hope that it is much better now. “The literature review on the study of natural sources of medicinal and biologically active compounds – nutraceuticals, is very extensive. Numerous studies concern not only plants (herbal medicine, use as food) but also fungi, including lichens and algae, bacteria and cyanobacterial [28]. Of particular interest are joint studies on several groups of organisms. In the available databases we have not been able to find a combined description of fungi (mushrooms), lichens, and algae in one paper, therefore we propose such a review. For example, in Poland, paradoxically, despite the tradition of collecting fungi for food (mycophilia), there is a lack of tradition and awareness of their use for medicinal purposes. Similarly when it comes to algae and lichens. In spite of numerous articles on the subject, there is still a lack of data, and mycological material, including lichenological and phycological, still represents a great potential for the protection of human health and requires further intensive research and increased awareness of how valuable these sources are. It is very interesting that these organisms used in ethnic medicine for thousands of years are valuable sources of biologically active substances, which is confirmed by contemporary research.” |
Besednova et al. Antiviral Effects of Polyphenols from Marine Algae. 2021;9(2):200.
|
Thank you for this comment. I added new reference to manuscript and new information about polyphenols in 5.4. section: “Polyphenols are heterogeneous group of compounds contains many biopolymers with unique structure and biological properties found in seaweed at high concentrations [169,187]. Algae (green, brown and red algae) accumulate phenolic compounds such as bromophenols, phenolic acids, flavonoids, phloroglucinol and its polymers such as phlorotannins [187,188]. Terrestrial plants accumulate only 2–3% of polyphenols while in Fucus reaching 3–12% of dry weight and in Ascophyllum nodosum 14% [187]. The largest amount of phlorotannins, accumulated within the vegetative cells of the outer cortical layer of the thalli, in Japanese Laminariaceae (Eisenia bicyclis, Ecklonia cava, Ecklonia kurome) was investigated [156,169]. Besednova et al. [187] noted that concentration of polyphenols in algae was significantly depending on the species, distributions, tissue type, water salinity, amount of nutrients, and extraction methods. Extracts from seaweeds containing polyphenolic metabolites are bioavailable, non-toxic to eukaryotic cells and, at the same time, have a pronounced bactericidal or bacteriostatic effect on a wide range of pathogenic microorganisms [187]. Variety of types of biopolymers and multifunctional properties (antioxidant, antiviral, antithrombotic, fungicidal, neuroprotective and antitumor activities) using the high potential for the medicinal application of polyphenols”. |
Mišurcová et al. Health benefits of algal polysaccharides in human nutrition. Adv Food Nutr Res. 2012;66:75-145.
|
Thank you for this comment. I added a new reference and more information about sulfate polysaccharides in 5.2. Polysaccharides section, I hope that I understood this comment correctly. In 5.2 Polysaccharides section: “Sulfate polysaccharides (containing hemiester sulfate groups in their sugar residues) exhibit immunomodulatory, antitumor, antithrombotic, anticoagulant, antimutagenic, anti-inflammatory, antimicrobial, and antiviral activities including anti-HIV infection, herpes, and hepatitis viruses [170]. Red algae (Chondrus crispus, Porphyra tenera, Schizymenia binderi) produce sulfated galactans consistin of the β-galactose or α-galactose units which have their anticoagulant activities. Mišurcová et al. [170] reported that 2,3-di-o-sulfated D-galactan from Botryocladia occidentalis exhibited anticoagulant activity comparable to heparin.” |
The authors also concluded by themselves “In recent years, polysaccharides with immunoregulatory and anticancer properties contained in some fungi and algae have been intensively researched” in their conclusion section.
|
Thank you for this comment. From medicinal point of view, especially sulfate polysaccharides are an important source of bioactive natural compounds exhibiting anticoagulant, antithrombotic, antitumor, antimicrobial, antimutagenic, anti-inflammatory, immunomodulatory, and antiviral activities. The results of many studies suggest that algal sulfated polysaccharides have a promising potential to be used as anticoagulant agents and medication for thrombotic disorders. |
Furthermore, some review papers had summarized the functions of bioactive components in fungi mushrooms. Such as Shahidi &Rahman reported the bioactives in seaweeds, algae, and fungi and their role in health promotion (J. Food Bioact. 2018;2:58-81). |
Thank you for this comment. This reference is under number 169 and it is an important paper about seaweeds, algae and fungi as rich sources of bioactive compounds. I added more information in 5.1. Carotenoids section: “Algae produce in addition to fucoxanthin other carotenoids such as astaxanthin, lutein and zeaxanthin. Canthaxanthin is synthesized by Chlorella vulgaris, Haematococcus pluvialis and shows antioxidant, anti-cancer, anti-diabetic activity. It also, prevents obesity [169]. I added also references in Table 3. |
Taken together, I believe it is really a hard work to summarize this review paper. But the beneficial of fungi and algae on human health has been recognized for a long time, may be not a new issue for the readers. |
Thank you for this comment. I added in Introduction section: “The literature review on the study of natural sources of medicinal and biologically active compounds – nutraceuticals, is very extensive. Numerous studies concern not only plants (herbal medicine, use as food) but also fungi, including lichens and algae, bacteria and cyanobacterial [28]. Of particular interest are joint studies on several groups of organisms. In the available databases we have not been able to find a combined description of fungi (mushrooms), lichens, and algae in one paper, therefore we propose such a review. For example, in Poland, paradoxically, despite the tradition of collecting fungi for food (mycophilia), there is a lack of tradition and awareness of their use for medicinal purposes. Similarly when it comes to algae and lichens. In spite of numerous articles on the subject, there is still a lack of data, and mycological material, including lichenological and phycological, still represents a great potential for the protection of human health and requires further intensive research and increased awareness of how valuable these sources are. It is very interesting that these organisms used in ethnic medicine for thousands of years are valuable sources of biologically active substances, which is confirmed by contemporary research”. |

Reviewer 2 Report
The quality of the manuscript is improved as result of the revision. However, the manuscript needs major revisions as outlined below:
- Table 1: what is the message for the reader? The nature of molecules, their structures and functions, or their applications? Moreover, I don’t understand the link between DNA/RNA, Proteins (column 1) and Mono-/Di-/Polysaccharides (column 2), as well as the link between Mono-/Di-/Polysaccharides and the functions reported in the column 3.
- Methodology: what keywords were used for the literature search?
- Line 138: please change “b-glucans” with “beta-glucans”
- Lines 145-150: what is the content in n-3 and n-6 PUFAs and what is their ratio?
- Lines 219-226: what are the nutritional properties of algae?The title of the paragraph is: “Nutritional value of edible mushrooms, and lichens and algae”. Authors describe some aspects related to the nutritional properties of algae in lines 664-673, but in that paragraph, Authors should discuss other aspects related to bioactive compounds!
- Line 622: according to the title of paragraph 4, paragraph 5 should be entitled: Algae bioactive compounds and their health-promoting properties.
- Lines 664-673: Authors should discuss other aspects related to bioactive compounds and not the aspects related to nutrients.
- A final figure summarizing the main messages of the study could be useful to the reader.
Author Response
Answers to points raised by the Reviewers of manuscript ID: nutrients-1368848
"Fungi and Algae as Sources of Medicinal and Other Biologically Active Compounds: A Review” by Joanna Åšlusarczyk et al.
I would like to thank you very much for the remarks about our manuscript titled “Fungi and Algae as Sources of Medicinal and Other Biologically Active Compounds: A Review”.
I am grateful to Reviewers for their comments. These comments allowed me to improve the quality of the manuscript. Below, I listed my answers to all questions and points raised by the Reviewers. All changes in revised manuscript are marked in red font.
Reviewer 2 |
|
Table 1: what is the message for the reader? The nature of molecules, their structures and functions, or their applications? Moreover, I don’t understand the link between DNA/RNA, Proteins (column 1) and Mono-/Di-/Polysaccharides (column 2), as well as the link between Mono-/Di-/Polysaccharides and the functions reported in the column 3. |
Thank you for this comment and I apologize for the confusion with table 1. I deleted Tables 1 in main manuscript and corrected number Table 2 and Table 3. I hope that it is much better now. |
Methodology: what keywords were used for the literature search?
|
Thank you very much for comment. I added in Methodology section: “Keywords such as pharmaceutical sources, biologically active compounds, medicinal mushrooms, edible mushrooms, functional foods, algae, fungi, lichens, nutraceuticals, ethnomycology, ethnolichenology, ethnomedicine were used to search the literature resources. Besides, information on particular species of fungi, lichens and algae was searched for health-promoting properties by entering their Latin names, also in combination with use and importance to people. The articles were also filtered for the presence of biologically active compounds such as phenolic compounds, flavonoid, carotenoid, polysaccharides, indole, sterols and vitamin in fungi, lichens and algae, as well as properties of these compounds such as anticancer, antioxidant, neuroprotective etc. and their nutritional characteristics”. |
Line 138: please change “b-glucans” with “beta-glucans” |
Done. |
Lines 145-150: what is the content in n-3 and n-6 PUFAs and what is their ratio? |
Thank you for your comment. I added the content in n-3 and n-6 PUFAs, as suggested. I hope that it is much better now. I added in 3. Nutritional value of edible mushrooms, lichens and algae section: “Percentage of fatty acids (in 100 g of total fatty acids) in mushrooms varies greatly: linoleic acid ranges from 0.0-0.81.1%, oleic acid between 1.0 and 60.3%, and linolenic acid from 0.0-28.8%. The consumption of essential fatty acids in balanced proportions (1:1 or 2:1 n-6/n-3) prevent obesity as an unbalanced n-6/n-3 ratio has been associated with adipogenesis. Also, essential fatty acids participate in high density lipoprotein formation which carries fat from the blood to the liver and reducing the risk of cardiovascular disorders [35]. The n-3 to n-6 ratios is (28.45±7.38% in cultivated edible compared to 16.19±7.38 to 55.42±9.03% in the wild medicinal mushrooms) higher than the recommended value of 2:1 or 3: 1 in human diets”. |
Lines 219-226: what are the nutritional properties of algae? The title of the paragraph is: “Nutritional value of edible mushrooms, and lichens and algae”. Authors describe some aspects related to the nutritional properties of algae in lines 664-673, but in that paragraph, Authors should discuss other aspects related to bioactive compounds! |
Thank you for your comments. I apologize for the confusion with the title of the paragraphs. I modified all paragraphs according to suggestion. I added: “Algae nutritional value increasingly are being presented as “functional foods” or “nutraceuticals” for describe foods that contain bioactive compounds, or phytochemicals, that may benefit health beyond the role of basic nutrition (e.g. anti-inflammatories, disease prevention). Protein content differs widely across groups of algae and various commercial species of the unicellular green alga Chlorella contain up to 70% dry weight protein [59,60]. Among the marine algae, red and green algae such as Porphyra spp. (laver), Pyropia spp. (nori), Palmaria palmata (dulse) and Ulva spp. (sea lettuce) often contain high levels of protein in contrast to lower levels in most brown algae. In amino acid composition in marine algae, glutamic acid, and aspartic acid represent the highest proportions of amino acids. These amino acids occur as protein constituents and as free amino acids or their salts. Glutamic acid content are decrease after several successive harvests of Pyronia yezoensis. Other amino acids (alanine and glycine) also contribute to distinctive flavors of some marine algae [61]. Marine macroalgae do not exceed 2-4.5% dry weight as lipids, mainly as phospholipids and glycolipids [60]. Lipid membranes contain sterols such as fucosterol and beta-sitosterol that also have reported health benefits. Fucosterol occurs in many algae, especially red and brown macroalgae, and it using in treating complications of diabetes and hypertension. Algal polysaccharides are the most widely, and often unknowingly, consumed food of algal origin. Edible macroalgae contain high amounts of dietary fiber, ranging from 23.5% (from Codium reediae) to 64.0% of dry weight in Gracilaria spp. [62]. Microalgal genera commonly considered as beneficial dietary supplements include Chlorella, Odontella, Porphyridium and Scenedesmus, with species Chlorella being recognized as particularly rich in polysaccharides. This algal bioactivity includes anticancer properties, cytokine modulation, ani-inflammatory effects, macrophage activation, and inhibition of protein tyrosine phosphatase. Acidic polysaccharide extracts from Chlorella pyrenoidosa have been patented as potentially useful antitumor and immunostimulating supplements [59,61]. Alginate is the major polysaccharide of brown algae, comprising 14-40% of its dry mass and was isolated as algin from kelp (Laminaria sp.). The abundant, heavily sulphated ulvans are extracted from members of the Ulvales. Bioactivities of ulvan extracts in vitro include antibacterial, anticoagulant, antioxidant, antiviral, antitumor and antihyperlipidemic effects. The main polysaccharides after the alginates in brown algae include beta-glucans (laminarans), cellulose, and heteroglycans”. |
Line 622: according to the title of paragraph 4, paragraph 5 should be entitled: Algae bioactive compounds and their health-promoting properties.
|
Thank you very much for comment. I corrected the title of paragraph 5 in main text in manuscript according to suggestion. Paragraph 5 is titled: Algae bioactive compounds and their health-promoting properties. |
Lines 664-673: Authors should discuss other aspects related to bioactive compounds and not the aspects related to nutrients. |
Thank you very much. I added information according to suggestion. I hope it is much better now. “Microalgal genera commonly considered as beneficial dietary supplements include Chlorella, Odontella, Porphyridium and Scenedesmus, with species Chlorella being recognized as particularly rich in polysaccharides. This bioactivity of algae contains anticancer properties, cytokine modulation, ani-inflammatory effects, macrophage activation, and inhibition of protein tyrosine phosphatase. Acidic polysaccharide extracts from Chlorella pyrenoidosa have been patented as potentially useful antitumor and immunostimulating supplements [59,61]. Alginate is the major polysaccharide of brown algae, comprising 14-40% of its dry mass and was isolated as algin from kelp (Laminaria sp.). The abundant, heavily sulphated ulvans are extracted from members of the Ulvales. Bioactivities of ulvan extracts in vitro include antibacterial, anticoagulant, antioxidant, antiviral, antitumor and antihyperlipidemic effects. The main polysaccharides after the alginates in brown algae are beta-glucans (laminarans), cellulose, and heteroglycans.” |
A final figure summarizing the main messages of the study could be useful to the reader. |
Thank you very much for comment. I added graphical abstract according to suggestion. I hope that it is much better now. |
With many thanks for all comments.
Sincerely,
Joanna Czerwik-Marcinkowska

Round 2
Reviewer 1 Report
The authors has modified the manuscript according the suggestions of reviewers.
Reviewer 2 Report
The Authors have addressed all my concerns and I have no further comments.
As far as I am concerned, the manuscript is now acceptable to be published.
This manuscript is a resubmission of an earlier submission. The following is a list of the peer review reports and author responses from that submission.
Round 1
Reviewer 1 Report
The authors of the review entitled “Algae and Fungi as Sources of Pharmaceuticals and Other Biologically Active Compounds” reviewed the medical application of derived compounds from mushrooms, lichens, and algae.
I consider that this revision must be revised in its structure and the authors should define a clear message for the readers.
It is not clear if the authors divided the sections depending on the function or depending on the proceeding compounds from fungi/lichens/algae.
In general, the manuscript was written in some confused manner and it is not clear the final message.
Comments:
Table 1: the function of the “Secondary metabolites” (Lipids, Fatty acids, etc.) lacks and the references are necessary.
Lines 26-30. The sentence was repeated.
Line 115. You already defined EFAs
Lines 130 – 134. Please, decide if you want to use the abbreviations of the elements or not.
Lines 132, 133. Please, provide de scientific name of the shiitake mushroom.
Lines 295 – 299. Please, add the reference.
Table 3: The references are necessary.
Table 4 and 1 are the same.
Lines 377 – 379. Add the references.
The references should be from international scientific journals.
Please, revise all references.
For example, the authors wrote: “The polysaccharides produced by shiitake […] are used in chemoprevention and as an adjunct in cancer therapy”. In my opinion, this is an important affirmation and it not be supported by the following reference: [9] TurÅ‚o, J. Biotechnologia grzybów. Zastosowanie w farmacji i suplementacji. Biuletyn WydziaÅ‚u Farmaceutycznego WUM. 2013, 3, 696 18–26.
The conclusion section is very large.
Reviewer 2 Report
Åšlusarczyk et al reported the latest scientific reports on bioactive selected substances contained in fungi and algae.
The topic of the review is interesting because Authors show that natural organisms could be used as supplementary medicine in the form of pharmaceutical preparations.
However, the manuscript often lacks clarity and needs some revisions.
1) Table 1, 3 and 4 could be organized in a single table in the Introduction section. This new table could be useful to the reader also to compare the composition of fungi and algae.
Moreover, the titles of columns (“Rank/Occurrence” and “Biogenesis/Structure”) are incorrect. Do DNA and RNA have an energetic function?
My suggestion is to introduce a new table reporting the names of molecules (classified as caloric nutrients, acaloric nutrients and bioactive molecules) present in fungi and/or algae and their supposed role for human health.
2) Paragraphs 2-5 look more like a list of information than a critical analysis of the literature: Authors just enunciate work made by others, instead of critically analysing it.
Moreover, paragraphs should be reorganized in a logical order. For example, “compounds with antioxidants properties” are also “compounds with anticancer properties”.
My suggestion is to critically describe the role of single molecule classes (caloric nutrients, acaloric nutrients, antioxidant molecules, other bioactive molecules, etc.) in human health, in terms of nutritional value and/or pharmaceutical interest.
A table summarizing results of the main literature studies could be added.
3) Paragraph 6 should be rewritten following the same logical order of previous paragraphs.
A table summarizing results of the main literature studies could be added.
Reviewer 3 Report
This manuscript summarized the nutritional value, active compounds, structural transmission, and biological benefits of algae and fungi, and are used as the sources of Industrial, pharmaceuticals, health-promoting properties and other biologically applications. Especially focus on mushrooms, lichens, and algae. It is really a hard work.
However, as the conclusions of authors, medicinal mushrooms, lichens and algae are a source with abundant biologically active substances, and many of which have been currently used in medical therapies. I suggest the authors should try to integrate more organized, to help the readers to getting the new and interesting point.